# Longitudinal observational cohort study: Speech for Intelligent cognition change tracking and DEtection of Alzheimer's Disease (SIDE-AD)

Stina Saunders [1], Fasih Haider,[2] Craig W Ritchie,[3,4] Graciela Muniz Terrera,[5,6] Saturnino Luz [7]

For numbered affiliations see end of article.

**Correspondence to**
Dr Stina Saunders;
Stina.Saunders@ed.ac.uk

## ABSTRACT

**Introduction** There is emerging evidence that speech may be a potential indicator and manifestation of early Alzheimer's disease (AD) pathology. Therefore, the University of Edinburgh and Sony Research have partnered to create the Speech for Intelligent cognition change tracking and DEtection of Alzheimer's Disease (SIDE-AD) study, which aims to develop digital speech-based biomarkers for use in neurodegenerative disease.

**Methods and analysis** SIDE-AD is an observational longitudinal study, collecting samples of spontaneous speech. Participants are recruited from existing cohort studies as well as from the National Health Service (NHS) memory clinics in Scotland. Using an online platform, participants record a voice sample talking about their brain health and rate their mood, anxiety and apathy. The speech biomarkers will be analysed longitudinally, and we will use machine learning and natural language processing technology to automate the assessment of the respondents' speech patterns.

**Ethics and dissemination** The SIDE-AD study has been approved by the NHS Research Ethics Committee (REC reference: 23/WM/0153, protocol number AC23046, IRAS Project ID 323311) and received NHS management approvals from Lothian, Fife and Forth Valley NHS boards. Our main ethical considerations pertain to the remote administration of the study, such as taking remote consent. To address this, we implemented a consent process, whereby the first step of the consent is done entirely remotely but a member of the research team contacts the participant over the phone to consent participants to the optional, most sensitive, elements of the study. Results will be presented at conferences, published in peer-reviewed journals and communicated to study participants.

## STRENGTHS AND LIMITATIONS OF THIS STUDY

⇒ The main strength of this study is that it combines assessment of cognition with mood, anxiety and apathy assessment over time, through spontaneous speech, which can be collected remotely, unobtrusively and at scale.

⇒ Another strength is its ability to relate patterns of speech collected while the participant is talking about their brain health to cognitive, clinical and functional outcomes through existing cohort data and National Health Service patient record data (optional consent).

⇒ Assessment by analysis of spontaneous speech, by relying on an inherently repeatable task that is not prone to practice effects, provides a way of avoiding variability in cognitive assessment due to extraneous environmental or subjective factors, which could distort the true trajectory of cognitive performance over time.

⇒ A limitation of the study is that there is no way to control for variability in audio recording equipment, such as background noise, as participants will be using their own personal computers or mobile phones to collect the speech samples.

⇒ Another possible limitation is that although the study will collect a standardised self-assessment baseline, there may be heterogeneity of data between the recruitment sources, which complicates the analysis.

## INTRODUCTION

In the current study, we will be analysing short recordings of individuals' speech to detect early changes indicative of cognitive decline in the older adult population. The following section contextualises the emergence of speech-based 'digital biomarkers' in the neurodegenerative disease area.

Dementia is a major public health concern with over 55 million people currently living with dementia worldwide and recent projections reaching 78 million living with dementia by 2030.[1] Alzheimer's disease (AD) is the most common underlying pathology for the dementia syndrome, accounting for 60% to 70% of all dementia cases[2,3] and is also one of the leading causes of death in the USA.[4] Due to a long preclinical period in the AD pathology, dementia is a late-stage manifestation of illness that has been developing years before dementia manifests.[5,6] Though not yet

globally available, recent pharmacological advances have yielded several new medications for the treatment of AD[7 8] and there are many other pharmacological approaches in progress, which necessitate detection of AD in its earliest pathological stage.

Dementia is an illness that causes neurodegeneration, whereby the brain gradually transitions from a healthy brain to brain failure.[9] In the initial symptomatic stages, deterioration of brain health manifests as cognitive impairment, though over time once dementia develops, the cognitive impairment cascades into problems with daily functioning. Additionally, symptomatic AD may change an individual's language, especially certain elements in speech such as elaboration (eg, making an argument clearer or providing examples) and attribution (eg, the ability to attribute characteristics to others).[10] What is more, evidence suggests that there is measurable difference between the speech patterns for individuals who are ageing with and without cognitive decline. For example, AD has been associated with shorter word lengths, higher pronoun-to-noun ratio and reduced use of nouns[11]; reduced meaningful content expressed, a reduction of total output of expressive language and a preference for generic verbs and simple syntax.[12] Acoustic features related to prosody and speech rhythms have also been found to be good predictors of AD.[13 14]

The significance of speech-based markers relates both to research and clinical practice. For instance, though the primary outcomes measured in AD clinical trials focus on cognition and biomarker data such as brain imaging, analysing cerebrospinal fluid or blood tests,[15] speech analysis could become a promising tool for detecting early neurodegenerative disease.[16] A systematic review of studies looking to distinguish healthy ageing from cognitive decline in older adults using speech features concluded that most studies record a diagnostic accuracy over 88% for AD and 80% for MCI.[13] Furthermore, a study by Eyigoz et al[17] predicted future diagnosis of AD among a cognitively normal baseline of Framingham Heart Study participants and demonstrated that linguistic features from a single administration of the cookie-theft picture description task out performed predictive models that incorporated genetic risk status, demographic variables and neuropsychological test results. Though limitations around specificity to underlying aetiology remain,[18] the ease of remote data collection makes speech biomarkers a valuable area to explore.

In our own systematic review on the potential of using interactive AI methods to detect AD biomarkers, we concluded that when compared with traditional neuropsychological assessment methods, speech and language technology were at least equally discriminative between different groups.[14] Interestingly, speech-based markers have been found to detect AD pathology such as amyloid-β status,[19] but evidence also suggests that people with higher amyloid levels (both asymptomatic and those with mild cognitive impairment) exhibit a greater rate of decline in episodic memory and language,[20] suggesting

the causal relation between speech and AD pathology should be explored further. Some of the most promising AD markers in speech may be lexico-semantic,[21] meaning the use of deictic words such as those referring to specific time (eg, now, tomorrow), place (eg, here, at home) or person (eg, I, the person). While research on speech biomarkers is only beginning to emerge, we expect speech and language analysis to play a role in detecting and tracking preclinical AD in the near future.[22 23]

In addition to cognition, speech and language are indicators of neurobehavioral outcomes such as mood, apathy and anxiety and have been used as input data in machine learning-based emotion recognition models.[24] Automatic methods for assessment of mood, anxiety and apathy could be an important component in AD detection and on-going monitoring of patient outcomes. Apathy is the most common neuropsychiatric symptom in patients with AD, and its accurate detection is important in terms of treatment outcomes, carer education and well-being.[25] Furthermore, mood disorders have also been linked to AD pathology.[26] However, as highlighted above, one of the limitations in speech biomarkers is that they may indicate deviations from typical function but not map to a specific neurodegenerative disease.[27]

To date, studies of neurodegenerative diseases involving speech have predominantly been laboratory based, and rarely included spontaneous speech. However, there are advantages in collecting spontaneous connected speech. Spontaneous speech can be captured in more natural settings over time, thus mitigating problems that might affect performance in controlled, cross-sectional data, such as a participant having an 'off day' or having slept poorly the night before the test. Speech can also be collected remotely, using widespread mobile devices and smartphones, which facilitates data collection at scale and over time.[28] The above practical considerations are critical as early recognition and intervention in AD may yield the most benefit.

This study explores changes in speech as potential early markers of pathological changes in AD. The Speech for Intelligent cognition change tracking and DEtection of Alzheimer's Disease (SIDE-AD) study is a longitudinal cohort study of individuals across the AD risk spectrum, those who are healthy volunteers and those engaged with secondary services for cognitive decline. The original contribution of our research is collecting speech data for the explicit analysis of AD speech markers, using connected speech as the most evidence-based modality for capturing relevant markers.

The study aims to develop digital speech-based biomarkers for AD in order to assess disease status in the predementia stages in a real-world population.

## METHODS AND ANALYSIS

This is a remote observational longitudinal study developing speech biomarkers for use in neurodegenerative disease. The SIDE-AD study involves collecting voice data

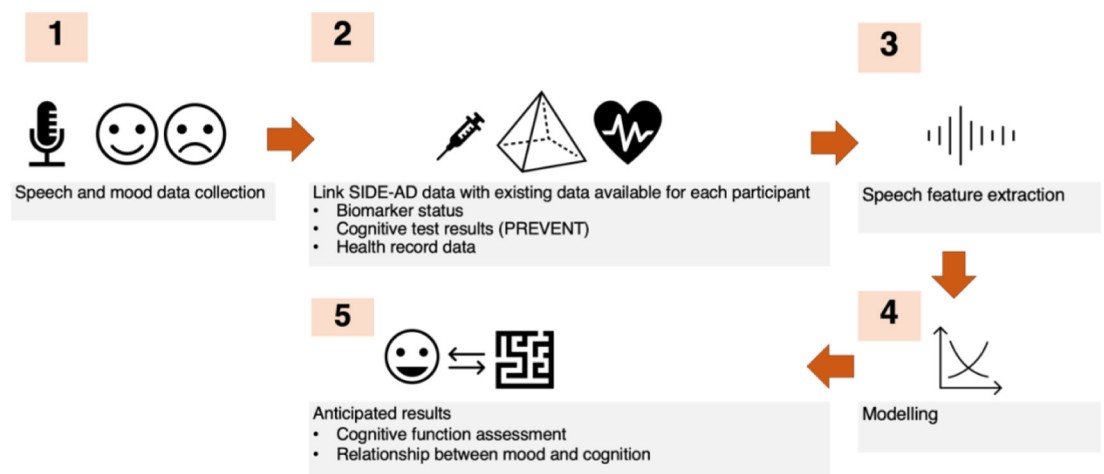

**Figure 1** Speech for Intelligent cognition change tracking and DEtection of AD (SIDE-AD) study flow.

using an online platform and prompting individuals to talk about brain health (figure 1), using protocol V.4, dated 29 November 2023.

## How the sample will be selected?

The selection of the participant population aims to create a risk spectrum of individuals with a low to high risk of AD. Therefore, the participant population includes individuals taking part in a longitudinal cohort study aimed at dementia prevention who are healthy volunteers but may have risk factors for dementia (such as genetic risk, increased amyloid or objective cognitive decline), individuals signed up for research registers in neurodegenerative disease and patients referred to clinical services who present with concerns around cognitive decline in National Health Service (NHS) memory clinics in the UK.

Participants will be identified through the different routes:

1. Participants taking part in the PREVENT Dementia study[29]

   The PREVENT study is a longitudinal cohort study of individuals in mid-life with n=700 participants, creating a spectrum of individuals from low to high risk of dementia. The participant information sheet (PIS) will be emailed by the PREVENT study team.

2. Participants will be identified through research registers such as Join Dementia Research, Scottish Brain Health Register, Scottish Health Research Register, the permission to contact list from the Scottish Dementia Research Interest Register held by the Neuroprogressive and Dementia Network (NDN).

   The PIS (SIDE-AD study PIS) will be emailed to all eligible participants who have expressed an interest towards the SIDE-AD study.

3. Participants will be identified through NHS memory assessment services

   Individuals will be either emailed the PIS electronically (online supplemental appendix 1) or handed the study PIS during routine clinic visits.

In order to achieve a spectrum of low to high individuals at risk of AD, additional recruitment sources may be added to the SIDE-AD study through protocol amendments. Protocol amendments will be communicated to relevant parties in line with regulatory requirements.

### Eligibility
#### Inclusion criteria
▶ Over 40 years old.
▶ Ability to carry out remote assessments (access to a computer/smart device).
▶ Have the capacity to consent.
▶ Participating in the PREVENT study or attending an NHS memory clinic.

#### Exclusion criteria
▶ Not able to speak English.
▶ Individuals who have not provided consent.

The SIDE-AD study uses an age limit of 40 years and older to be consistent with the inclusion criteria in the PREVENT Dementia study eligibility criteria.

### Sample size calculation
We aim to recruit a total of 450 participants over a 10-month period (October 2023–August 2024). The PREVENT study has 700 participants overall and at the first stage, only participants from Scotland (n=300) will be invited to take part in the SIDE-AD study. Additionally, we estimate the number of eligible individuals in NHS clinical settings to be around 150 over 10 months, which will be sufficient for our planned analyses.

There is no a priori sample size calculation as the SIDE-AD study will recruit from the PREVENT study, research registers and enrich the sample with a clinical population recruiting from NHS memory clinics where all eligible individuals will be invited to take part in the SIDE-AD study. We used Riley *et al's*[30] approach to calculate the sample size and elaborate on the process below.

It is difficult to estimate precisely the optimal sample sizes and measurable effect sizes for machine learning-based studies such as SIDE-AD. In the field of medical decision support systems, a commonly adopted rule of thumb for determining sample size in continuous outcome settings is to require two participants per feature (predictor) in the model. While this heuristic approach has been criticised for not taking into account the context of different studies,[30] some theoretical justifications for it have been offered in the pattern recognition literature for the case of classification.[31] According to this approach, for relatively simple machine learning algorithms such as Euclidean distance and linear discriminant analysis, lower bounds at 1.2 and 1.4 participants per feature have been proposed, for an expected probability of misclassification (PMC) at most 50% greater than an asymptotic PMC of 0.1. Using the enhanced Geneva Minimalistic Acoustic Parameter Set (eGeMAPS) feature set[32] as proposed here would imply a minimum of 114 participants for a PMC at most 50% greater than a conservative asymptotic PMC of 0.1 (90% accuracy). However, this estimate presupposes the use of a fairly simple linear model. We aim to collect data from larger numbers of participants, which will allow us to experiment with larger sets of speech features. In Riley *et al*'s paper,[3] a method is proposed for estimation of sample sizes for continuous outcome regression models that aim to ensure that the data sample is large enough to estimate the model intercept and residual variance precisely, and account for the risk of overfitting by setting adequate regularisation and optimism targets. While this approach only covers standard regression models, it provides more conservative sample size estimates. Taking a previous study on anxiety and depression recognition from speech conducted by our group[33] as a basis, and assuming a model $R^2$ of 0.3, a regularisation factor of 0.8, mean arousal and valence scores of 30, standard deviation (SD) of 22 and a reduced feature set of 40 features would require sample size of 421 participants, according to the formula given in Riley *et al*.[30] It is speculated[30] that machine learning models require much larger sample sizes than simpler statistical models. We intend to use existing datasets containing speech samples from over 1000 subjects for data augmentation and dimensionality reduction methods, such as Haider *et al*[34] should our study's sample size prove insufficient to train accurate predictive models.

## What outcomes will be measured?

As this study aims to develop models for prediction of AD and related neurobehavioural outcomes, the primary endpoints of this study are measures of model prediction performance, namely, root mean squared error (RMSE) and coefficient of determination values for predicted cognitive scores, mood, anxiety and apathy prediction ratings, for regression models and area under the receiver operating characteristic curve (ROC), sensitivity and specificity for classification into score categories (eg, high vs low mood, etc).

**Table 1** Variables collected as part of SIDE-AD study

| | |
|---|---|
| Spontaneous speech | Voice-recording in response to: 'Please tell us in your own words how you look after your brain health'. |
| Sociodemographic details | Date of birth<br>Gender<br>Education |
| Self-reported neurodegenerative disease diagnosis | Mild cognitive impairment<br>Alzheimer's disease<br>Mixed dementia<br>Dementia with Lewy bodies<br>Frontotemporal dementia<br>Vascular dementia<br>Parkinson's disease dementia<br>Parkinson's disease<br>Huntington's disease<br>Creutzfeldt-Jakob disease<br>Korsakoff syndrome<br>Motor neuron disease (also known as ALS or amyotrophic lateral sclerosis)<br>Subjective cognitive disorder<br>Major neurocognitive disorder<br>Stroke |
| Self-reported mood | 1–100 score |
| Self-reported anxiety | 1–100 score |
| Self-reported apathy | 1–100 score |

In terms of secondary endpoints, all participants in the SIDE-AD study will self-report the following outcomes, which are used as explanatory variables (table 1).

For participants recruited through the PREVENT Dementia study, SIDE-AD will have access to the following variables collected as part of the PREVENT protocol:

► Demographics (date of birth; gender; race; education).
► Biosamples (APOE e4 status, amyloid status).
► Risk factors (self-reported memory concerns and health status, being out of breath, family history of dementia).
► Lifestyle interview (smoking status, electronic cigarette use, drug use, alcohol intake, diet).
► Cognitive outcomes
   – National Adult Reading Test.
   – Visual Short-Term Memory Binding Test.
   – Cognito.
   – ACE-III.
   – Supermarket task.
   – 4 Mountains task.
► Clinical and functional outcomes
   – Apathy Inventory.
   – Lawton Instrumental Activities Of Daily Life Scale
   – Center for Epidemiological Studies Depression (CES-D) Scale.
   – Spielberger State Scale
   – Pittsburgh Sleep Quality Index
   – Sleep Questionnaire

- Epworth Sleepiness Scale
- Life Stressor Checklist
- Brain Injury Screening Questionnaire
► Dementia Risk Score
  - Cardiovascular Risk Factors, Aging and Dementia (CAIDE) score.

Individuals who are recruited through NHS services will have an optional consent for record linkage. For those participants who are recruited through NHS services and who consent to record linkage, changes in speech will be associated with routinely collected healthcare data.

Record linkage will include any/all the available outcomes:
► Neurodegenerative disease diagnosis
► Clinical outcomes (ADL, mood/anxiety and sleep measures)
► Neuropsychological outcomes (cognitive tests)
► Biomarker outcomes: genetic risk (ApoE4)
► Biomarker outcomes: amyloid levels
► Biomarker outcomes: τ levels
► Neuroimaging outcomes

We note that each of the above variables will be linked with the time stamp of when this was reported (eg, if a diagnosis is reported during one of the follow-up appointments, time since diagnosis is controlled for in the analysis).

### When and how?
The SIDE-AD study is self-administered remotely, though a member of the research team is available via contact details provided on the study PIS and study website. The assessment takes around 5 min to complete on a computer or smart device and the participant receives an email invitation for follow-up assessments every 3 months.

Interested participants follow a unique URL link to the SIDE-AD research platform, where they will be asked to give consent (online supplemental appendix 2) and perform the baseline assessment. This is done remotely from the participants' home with no direct involvement of the research team. All individuals in the SIDE-AD study will be invited to provide speech samples through the online platform. Assessments, including speech sampling, will be done quarterly over the study duration.

Participant retention and completion of follow-up are promoted through reminders in each follow-up invitation that participants' contribution is valued and voluntary.

### Data analysis plan
We will conduct descriptive analysis of the sociodemographic information provided by the individuals, using simple tabulations. A lay language report with the survey results from the analysis of the data will be produced and made publicly available.

We developed the SIDE-AD platform to collect speech data and limited background/self-reported mood, anxiety and apathy data. The platform is used to monitor speech patterns over time. We will extract speech markers

and evaluating a comprehensive set of acoustic features for modelling and prediction of cognitive function.

We will use the primary data for the purpose of developing models for the assessment of cognition, mood, anxiety and apathy. We will extract the following speech features and their most common statistical functionals, in total, 88 low-level descriptors of the eGeMAPS as per Eyben *et al*[32] from the collected speech signal.

► **F0 semitone:** F0 is determined physiologically by the number of cycles that the vocal folds make in a second, corresponding to prosodic aspects of speech.
► **Loudness:** the loudness of speech implies an increase in the vocal effort, which describes a strained or tense voice quality.
► **Spectral flux:** a low-level feature that measures how quickly the power spectrum of a signal changes, used for voice activity detection.
► **Mel-frequency cepstral coefficients** represent spectral information from the speech signal, using a scale known as the 'Mel-frequency scale', which is chosen to mimic the way humans perceive audio.
► **Jitter:** this indicates the variability or perturbation of F0 (ie, a measure of frequency instability).
► **Shimmer:** this is also a measure of F0 perturbation but is related to the amplitude of the sound wave or the intensity of vocal emission (ie, a measure of amplitude instability). A normal healthy voice has a small amount of instability during sustained vowel production, and tissue and muscle properties influence normal instabilities.
► **F1, F2, F3:** these are formant frequencies that relate to characteristics of the speaker's vocal tract.
► **Alpha ratio:** the ratio between the energy in the low-frequency region and the high-frequency region; a measure correlated to perceived voice quality.
► **Hammarberg index:** a spectral measure used as an energy distribution measure averaged across an utterance.
► **Spectral slope** (linear regression within bands 0–500 Hz and 500–1500 Hz): a feature that has been investigated in connection with emotional speech.

Once these features have been extracted, we will apply the active data representation method, Haider *et al*,[34] for dimensionality reduction, in order to extract the relevant dimensions of variation of the signal, while downplaying contingent voice features. Finally, we will train support vector regression models for automatic prediction of cognitive status (cognitive test scores), and mood, anxiety and apathy levels (self-reported). We will also explore the relationship between these inferred and self-reported neurobehavioral outcomes and cognitive function through Pearson correlation statistical test and through machine learning modelling, with mood, anxiety and apathy used as additional input features for the model, along with speech features. Where possible, we will discretise the cognitive and neurobehavioral outcomes (mood ratings can be assigned to three classes: depressed, healthy and (hypo)manic) and train classification models

to predict the corresponding classes. Performance of regression models will be assessed throughout the RMSE metric and the coefficient of determination. Performance of classification models will be assessed by computing the area under the ROC curve and sensitivity and specificity metrics. We will also generate speech transcripts from the recorded audio and extract word embeddings[35] and general linguistic features found to be useful predictors in previous studies and compare the performance of models built with these features with those based on acoustic features. As we are collecting spontaneous rather than task-based speech, we hypothesise that acoustic features will generalise better to the less constrained setting of our study.

### Ethics and dissemination: ethical and safety considerations
We consider the main issues to pertain to the remote administration of the study. We have included contact details of the research team on every page of the survey and invite individuals to contact the team if they have any questions after reading the PIS accompanying the informed consent form. We have also engaged with the study sponsor Academic and Clinical Central Office for Research and Development (ACCORD) and the NHS information governance and IT security teams from the initial design stages of the study protocol to ensure remote consent is a secure method for consent. Our solution was to implement a two-stage consent process whereby the first step of the consent is done entirely remotely but the participant has an option to consent to being contacted about linking to relevant sections in their memory clinic records. If participants consent to being contacted about record linkage, a member of the SIDE-AD study research team contacts the participant and while the second step of the optional record linkage is electronic, it is only after a conversation with a member of the research team, which also allows the research team to assess capacity to consent. We note that the optional record linkage consent only applies to a small number of participants who are recruited through NHS memory clinics.

We considered this to be an appropriate process from both Good Clinical Practice and data security point of view, but this was also the favoured method to consent by our patient and public involvement (PPI) contributors who gave feedback at the design stage of the protocol/survey. The feedback also considered the study to be fairly low impact only taking a few min to complete which does not necessitate face to face consent.

The other issue we have carefully considered is recording individuals' speech. As collecting speech data for clinical research can be difficult due to privacy issues,[36] we remind individuals not to disclose personally identifiable information in the speech recording and we also explain that recording of the speech sample is for research purposes only and not part of clinical care.

### The patient and public involvement statement
Patients and the public were first involved in the research at the design stage. We will also involve patients and public in the analysis of results and dissemination of findings. For the design stage, we contacted the NDN PPI panel in Scotland for feedback on our survey platform, PIS and consent forms. We received feedback from 10 individuals both healthy volunteers and those with lived experience of neurodegenerative disease. We carefully reviewed feedback and implemented changes proposed by the PPI contributors, these pertained not only to the online survey platform but also to the information provided about the study. Overall, the PPI input was favourable, and individuals saw value in a remote study developing speech-based biomarkers, which was perceived to have low participant burden but reward for taking part in research. Some of the issues raised involved giving more guidance on what people could talk about when they are asked to record speech and to add a box for optional comments for if individuals want to add a note about their recording to the research team. We also intend to involve PPI contributors in the analysis of results and dissemination of findings to help prioritise what information to share, at which conferences/workshops and in which format. We plan to do this via the NDN; EPAD participant panel or National Dementia Carers Action Network (part of Alzheimer Scotland).

### Dissemination plan
A possible dissemination opportunity will be the Scottish Dementia Research Consortium annual meeting. Additionally, we plan to publish findings from this study in peer-reviewed scientific journals, international conferences such as the AAIC, INTERSPEECH and ICASSP and communicate the study results to participants, healthcare professionals and the public by publishing a lay language report on the university website. We plan to publish a description of the data in venues like Nature Scientific Data and present it at the language resources and evaluation conference.

### Data storage
During the study, data will be securely stored on the University of Edinburgh's DataStore service, where it will be accessible only to the research team and not shared outside of the research team in order to protect confidentiality. No personal information about potential participants will be collected as study team will not have access to the individuals' contact details who have been disseminated information about the SIDE-AD study by their parent cohort or NHS clinical teams. Personal information about enrolled participants will be stored securely and separately from research data.

After the study, for longer term preservation of the project's legacy and to ensure reproducibility of our analyses, anonymised research data will be stored on the University of Edinburgh Datashare (https://datashare.ed.ac.uk/handle/10283/8553), managed by the University

Research Data Service, where the data will be assigned a digital object identifier. As this project collects voice data, it is not possible to anonymise the data entirely.

**Author affiliations**
¹University of Edinburgh, Edinburgh, UK
²Usher Institute, University of Edinburgh, Edinburgh, UK
³Department of Psychiatry, Centre for Clinical Brain Sciences, University of Edinburgh, Edinburgh, UK
⁴Scottish Brain Sciences, Edinburgh, UK
⁵Centre for Dementia Prevention, University of Edinburgh, Edinburgh, UK
⁶Heritage College of Osteopathic Medicine, Ohio University, Athens, Ohio, USA
⁷Usher Institute of Population Health Sciences and Informatics, University of Edinburgh School of Molecular Genetic and Population Health Sciences, Edinburgh, UK

**Acknowledgements** Thank you to the participants enrolled in the SIDE-AD study. We acknowledge Sony Research Awards for their funding of the project and thank the Sony team for their contribution to our thinking around this work.

**Contributors** SS created a first draft of the protocol with input from SL and FH. SL conceived the idea for the study. SL and SS conceived the design for the study. FH developed the survey software and data management and analysis' plan with input from SS and SL. SS completed the regulatory approvals with input from FH and SL. FH and SS also performed patient and public involvement. SS revised the protocol per suggestions from SL, CWR, GMT. All authors read and approved the final protocol and manuscript for the current paper.

**Funding** This work was supported by Sony Research Award grant number 11477634- c-00008809.This research was funded in part by UKRI project number 10102226. For the purpose of open access, the author has applied a creative commons attribution (CC BY) licence to any author accepted manuscript version arising.

**Competing interests** None declared.

**Patient and public involvement** Patients and/or the public were involved in the design, or conduct, or reporting, or dissemination plans of this research. Refer to the Methods section for further details.

**Patient consent for publication** Not applicable.

**Provenance and peer review** Not commissioned; externally peer-reviewed.

**Data availability statement** Data available on reasonable request. Interested parties can contact the Pprimary Iinvestigator for request for data and access can be provided on reasonable request with relevant regulatory guidance in place.

**ORCID iDs**
Stina Saunders http://orcid.org/0000-0003-3323-2505
Saturnino Luz http://orcid.org/0000-0001-8430-7875

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
