## [Reviewer comments · BMJ Open]

ARTICLE DETAILS

TITLE (PROVISIONAL)	Protocol: Longitudinal observational cohort study: Speech for Intelligent cognition change tracking and DETection of Alzheimer's Disease (SIDE-AD)
AUTHORS	Saunders, Stina; Haider, Fasih; Ritchie, Craig; Muniz Terrera, Graciela; Luz, Saturnino

VERSION 1 – REVIEW

REVIEWER	Ramanarayanan, Vikram University of California San Francisco
REVIEW RETURNED	20-Dec-2023

GENERAL COMMENTS	The authors propose a protocol for a longitudinal observational cohort study designed to detect and tracking progress of AD in a cohort of 700 participants. The manuscript reads well, and I would recommend the authors make minor revisions to clarify several points, stated below, and add information where needed. *References: The references can be made more comprehensive. I would recommend citing additional literature that captures the breadth of existing work in this field. Examples include: Boschi, V., Catricala, E., Consonni, M., Chesi, C., Moro, A., & Cappa, S. F. (2017). Connected speech in neurodegenerative language disorders: a review. Frontiers in psychology, 8, 269. Ramanarayanan, V., Lammert, A. C., Rowe, H. P., Quatieri, T. F., & Green, J. R. (2022). Speech as a biomarker: Opportunities, interpretability, and challenges. Perspectives of the ASHA Special Interest Groups, 7(1), 276-283. Voleti, R., Liss, J. M., & Berisha, V. (2019). A review of automated speech and language features for assessment of cognitive and thought disorders. IEEE journal of selected topics in signal processing, 14(2), 282-298. Robin, J., Xu, M., Balagopalan, A., Novikova, J., Kahn, L., Oday, A., ... & Teng, E. (2023). Automated detection of progressive speech changes in early Alzheimer's disease. Alzheimer's & Dementia: Diagnosis, Assessment & Disease Monitoring, 15(2), e12445. *pg 6, lines 25-27. Some other studies have also done this. How is this study similar to or different from those? - Questions about methods: *pg 7, lines 5-7. How did you come up with the N=700 number? Did you do a power analysis for sample size estimation or is this just an upper bound on the estimate number of subjects you
--

	forecast being able to recruit? Also, it wasn't clear to me how you will ensure that you get a uniform distribution of participants across age/sex and AD/dementia risk. *pg 7. Why did you pick 40 years old and higher? *pg 8-9, Table 1. Why are you only restricting your speech task to free spontaneous speech? Why not additional ones, such as a picture description task, which have been shown to be useful in prior publications? * Are you collecting years since diagnosis? Are you going to restrict people based on how many years have passed since they received their diagnosis? *pg 11, Features. In addition to acoustic features, such as the eGeMAPs set, is there any reason you aren't looking at interpretable linguistic/text features (which, again, have been shown to be useful in prior art?)
--	--

REVIEWER	Alamrawy, Roa Mamoura Psychiatric Hospital, General Secretariat of Mental Health and Addiction Treatment
REVIEW RETURNED	25-Dec-2023

GENERAL COMMENTS	Dear Authors, I am impressed by the originality and thoroughness of your research idea, as well as the clarity and organization of your writing. However, I suggest some minor points for improvement, which I believe can further strengthen your manuscript: Try to highlight what you are looking for earlier in the protocol, as in speech detection, try to clarify earlier, what you will looking for and what will be its meaning. Also, in rationale behind the research idea, try to speak more about the current knowledge in this area. Such topic is novel and you need to satisfy basic information first for your audience before expanding in such advanced area. Addressing these points will enhance the overall impact of your work and ensure its alignment with the high standards of BMJ. Thank you for your excellent work and your commitment, Sincerely, Dr. Roa Gamal Alamrawy, MD
--

REVIEWER	Novotny , Michal
REVIEW RETURNED	02-Jan-2024

GENERAL COMMENTS	Thank you for the opportunity to review a study protocol called Speech for Intelligent cognition change tracking and DEtection of AD (SIDE-AD) aimed at the computerized detection of Alzheimer's disease (AD) manifestations in speech. The authors plan to enroll 450 participants over 40 years of age participating in the PREVENT study or attending an NHS memory clinic with a low and high risk of AD development. Participants will provide samples of speech (talk about brain health) collected through an online platform. The data analysis will describe several acoustic features. The topic of the study is of great interest, especially in light of the search for an AD treatment for which the early markers may select candidates for clinical trials or preventive medicine. My concern for the study is that the selected features may, to a certain extent, reflect cognitive deficits; however, they are designed to detect motor deficits. I would suggest including the
--

	linguistic features used for assessing cognitive deficits in multiple sclerosis or synucleinopathy. Subert M, Novotny M, Tykalova T, Srpova B, Friedova L, Uher T, Horakova D, Rusz J. Lexical and Syntactic Deficits Analyzed via Automated Natural Language Processing: The New Monitoring Tool in Multiple Sclerosis. Ther Adv Neurol Disord 2023 §Subert M, Simek M, Novotny M, Tykalova T, Bezdicek O, Ruzicka E, Sonka K, Dusek P, Rusz J. Linguistic abnormalities in isolated rapid eye movement sleep behavior disorder. Mov Disord 2022;37:1872-1882.
--	--

VERSION 1 – AUTHOR RESPONSE

Reviewer Comments:

Reviewer #1: The authors propose a protocol for a longitudinal observational cohort study designed to detect and tracking progress of AD in a cohort of 700 participants. The manuscript reads well, and I would recommend the authors make minor revisions to clarify several points, stated below, and add information where needed.

Thank you for the positive comments on our work and the thoughtful recommendations to clarify certain points.

*References: The references can be made more comprehensive. I would recommend citing additional literature that captures the breadth of existing work in this field. Examples include:

Boschi, V., Catricala, E., Consonni, M., Chesi, C., Moro, A., & Cappa, S. F. (2017). Connected speech in neurodegenerative language disorders: a review. *Frontiers in psychology*, 8, 269.

We have cited the above paper by adding the following sentence (page 4): “Some of the most promising AD markers in speech may be lexico-semantic (Boschi et al., 2017), meaning the use of deictic words such as those referring to specific time (e.g., now, tomorrow), place (e.g., here, at home), or person (e.g., I, the person).”.

Ramanarayanan, V., Lammert, A. C., Rowe, H. P., Quatieri, T. F., & Green, J. R. (2022). Speech as a biomarker: Opportunities, interpretability, and challenges. *Perspectives of the ASHA Special Interest Groups*, 7(1), 276-283.

We have cited the above paper by adding the following sentence (page 5): “However, as highlighted above, one of the limitations in speech biomarkers is that they may indicate deviations from typical function but not map to a specific neurodegenerative disease (Ramanarayanan et al., 2022).”.

Voleti, R., Liss, J. M., & Berisha, V. (2019). A review of automated speech and language features for assessment of cognitive and thought disorders. *IEEE journal of selected topics in signal processing*, 14(2), 282-298.

We have cited the above paper by adding the following sentence (page 12): “As collecting speech data for clinical research can be difficult due to privacy issues (Voleti et al., 2019), we remind individuals not to disclose personally identifiable information in the speech recording and we also explain that recording of the speech sample is for research purposes only and not part of clinical care.”.

Robin, J., Xu, M., Balagopalan, A., Novikova, J., Kahn, L., Oday, A., ... & Teng, E. (2023). Automated detection of progressive speech changes in early Alzheimer's disease. *Alzheimer's & Dementia: Diagnosis, Assessment & Disease Monitoring*, 15(2), e12445.

We have cited the above paper by adding the following sentence (page 4): “For example, AD has been associated with shorter word lengths, higher pronoun-to-noun ratio, and reduced use of nouns (Robin et al., 2023); reduced meaningful content expressed, a reduction of total output of expressive language and a preference for generic verbs and simple syntax (Williams et al., 2021).”.

*pg 6, lines 25-27. Some other studies have also done this. How is this study similar to or different from those?

Thank you for raising this point, the speech data collected in our study will be linked to electronic health records which enables an analysis between speech and electronic health records data. We have added further context to highlight the novelty of our work in the manuscript. The section now reads (page 5): “The original contribution of our research is collecting speech data for the explicit analysis of AD speech markers, using connected speech as the most evidence-based modality for capturing relevant markers.”.

- Questions about methods:

*pg 7, lines 5-7. How did you come up with the N=700 number? Did you do a power analysis for sample size estimation or is this just an upper bound on the estimate number of subjects you forecast being able to recruit? Also, it wasn't clear to me how you will ensure that you get a uniform distribution of participants across age/sex and AD/dementia risk.

The n=700 refers to the total number of participants taking part in the PREVENT study (though at the first stage, only participants from Scotland will be invited to take part in the SIDE-AD study).

The recruitment aim of a total of 450 participants over a 10-month period (Oct 2023-Aug 2024) is based on a power calculation based on Riley et al. (2020) work. We added a clarification in the manuscript to highlight this (page 7): “We used the Riley et al. (2020) approach to calculate the sample size and elaborate on the process below.”.

We have carefully considered how we get a uniform distribution of participants across age/sex and AD/dementia risk. To achieve this aim, we recruit through a number of recruitment sources – a longitudinal cohort study with participants with and without family history of AD; research registers for neurodegenerative disease (assuming this source may be biased towards people with higher awareness and possible experience of supporting someone with dementia) and additionally, patients in NHS settings who will form the highest risk/diagnostic group in our sample. While we control for age through eligibility criteria, we note that it is difficult to control for sex of the participants a priori. However, all subsequent analysis will be controlled for sociodemographic factors to ensure any change observed is not secondary to age/gender/education.

*pg 7. Why did you pick 40 years old and higher?

We have added the following sentence (page 7): “The SIDE-AD study uses an age limit of 40 years and older to be consistent with the inclusion criteria in the PREVENT Dementia study eligibility criteria.”.

*pg 8-9, Table 1. Why are you only restricting your speech task to free spontaneous speech? Why not additional ones, such as a picture description task, which have been shown to be useful in prior publications?

We consider free spontaneous speech as the most optimal method for developing models for monitoring subjects through their naturalistic daily free speech. The picture description task in some ways is more intrusive to elicit spontaneous free speech in comparison to truly unstructured prompts.

* Are you collecting years since diagnosis? Are you going to restrict people based on how many years have passed since they received their diagnosis?

We are not restricting people based on number of years passed since diagnosis and have added the following sentence in the manuscript (page 10): “We note that each of the above variables will be linked with the time stamp of when this was reported (e.g., if a diagnosis is reported during one of the follow-up appointments, time since diagnosis is controlled for in the analysis).”.

*pg 11, Features. In addition to acoustic features, such as the eGeMAPs set, is there any reason you aren't looking at interpretable linguistic/text features (which, again, have been shown to be useful in prior art?)

Thank you for the suggestion to look at interpretable linguistic/text features. We focus on acoustic features due to the fact that the speech samples we will collect are spontaneous, unconstrained speech, as opposed to, say, picture descriptions, where the vocabulary and semantics are more tightly constrained. Linguistic features are advantageous in the latter case, as lexical, syntactic, and semantic features can be more easily compared across participants. We believe acoustic features will generalise better for the speech data we are targeting. We have however included linguistic features in our analysis plan and added the following sentence to clarify our approach further (page 12): “We will also generate speech transcripts from the recorded audio and extract word embeddings (Devlin et al 2018) and general linguistic features found to be useful predictors in previous studies and compare the performance of models built with these features with those based on acoustic features. As we are collecting spontaneous rather than task-based speech, we hypothesise that acoustic features will generalise better to the less constrained setting of our study.”.

Reviewer #2: Dear Authors,

I am impressed by the originality and thoroughness of your research idea, as well as the clarity and organization of your writing. However, I suggest some minor points for improvement, which I believe can further strengthen your manuscript:

Thank you for recognising the strength of our work, the positive feedback on the manuscript and the suggestions to improve the manuscript further.

Try to highlight what you are looking for earlier in the protocol, as in speech detection, try to clarify earlier, what you will looking for and what will be its meaning.

Thank you, we added a section to the start of introduction (page 3): “In the current study, we will be analysing short recordings of individuals' speech to detect early changes indicative of cognitive decline in the older adult population. The following section contextualises the emergence of speech-based “digital biomarkers” in the neurodegenerative disease area.”.

We also added (page 4) “The significance of speech-based markers relates both to research and clinical practice” and “Though limitations around specificity to underlying aetiology remain (Brzezińska et al., 2020), the ease of remote data collection makes speech biomarkers a valuable area to explore.”

Also, in rationale behind the research idea, try to speak more about the current knowledge in this area. Such topic is novel and you need to satisfy basic information first for your audience before expanding in such advanced area.

We believe the addition of evidence from the work suggested by Reviewer #1 adds to the current knowledge in this area. We also elaborated on the mixed evidence around speech and underlying AD pathology in the introduction section.

We revised the following sentence (page 4): “Interestingly, speech-based markers have been found to detect AD pathology such as amyloid- β status (Hajjar et al., 2023), but evidence also suggests that people with higher amyloid levels (both asymptomatic and those with mild cognitive impairment) exhibit a greater rate of decline in episodic memory and language (Lim et al., 2014), suggesting the causal relation between speech and AD pathology should be explored further.” and added the following sentence: “Acoustic features related to prosody and speech rhythms have also been found to be good predictors of AD (Martínez-Nicolás et al., 2021; de la Fuente et al., 2020).”.

Addressing these points will enhance the overall impact of your work and ensure its alignment with the high standards of BMJ.

Thank you for your excellent work and your commitment

Thank you for the valuable feedback.

Reviewer #3: Thank you for the opportunity to review a study protocol called Speech for Intelligent cognition change tracking and DEtection of AD (SIDE-AD) aimed at the computerized detection of Alzheimer’s disease (AD) manifestations in speech. The authors plan to enroll 450 participants over 40 years of age participating in the PREVENT study or attending an NHS memory clinic with a low and high risk of AD development. Participants will provide samples of speech (talk about brain health) collected through an online platform. The data analysis will describe several acoustic features. The topic of the study is of great interest, especially in light of the search for an AD treatment for which the early markers may select candidates for clinical trials or preventive medicine.

Thank you for considering the topic of our study interesting in light of new AD treatments.

My concern for the study is that the selected features may, to a certain extent, reflect cognitive deficits; however, they are designed to detect motor deficits. I would suggest including the linguistic features used for assessing cognitive deficits in multiple sclerosis or synucleinopathy.

Thank you for the suggestion. We have included linguistic features in our analysis plan and added the following sentence (page 12) to clarify our approach further: “We will also generate speech transcripts from the recorded audio and extract word embeddings (Devlin et al 2018) and general linguistic features found to be useful predictors in previous studies and compare the performance of models built with these features with those based on acoustic features. As we are collecting spontaneous rather than task-based speech, we hypothesise that acoustic features will generalise better to the less constrained setting of our study”.

Subert M, Novotny M, Tykalova T, Srpova B, Friedova L, Uher T, Horakova D, Ruz J. Lexical and Syntactic Deficits Analyzed via Automated Natural Language Processing: The New Monitoring Tool in Multiple Sclerosis. *Ther Adv Neurol Disord* 2023

§Subert M, Simek M, Novotny M, Tykalova T, Bezdicek O, Ruzicka E, Sonka K, Dusek P, Ruz J. Linguistic abnormalities in isolated rapid eye movement sleep behavior disorder. *Mov Disord* 2022;37:1872-1882.